# The burden of premature adult mortality associated with lack of access to electricity in India

Vittal Hejjaji[1], Dweep Barbhaya[2], Amirarsalan Rahimian[3], Aishwarya Yamparala[4], Shreyas Yakkali[5], Aditya K. Khetan[6]*

1 Department of Cardiovascular Medicine, Saint Luke's Mid America Heart Institute, Kansas City, Missouri, United States of America, 2 Department of Medicine, MedStar Washington Hospital Center, Washington, DC, United States of America, 3 Department of Medicine, Research Assistant, McMaster University, Hamilton, Ontario, Canada, 4 Department of Medicine, Andhra Medical College, Visakhapatnam, Andhra Pradesh, India, 5 Department of Medicine, Seth G.S. Medical College and K.E.M. Hospital, Mumbai, Maharashtra, India, 6 Department of Cardiovascular Medicine, McMaster University, Hamilton, Ontario, Canada

* khetana@mcmaster.ca

**Data Availability Statement:** All relevant data are within the manuscript and its Supporting information files.

## Abstract

### Background

The impact of electricity access on all-cause premature mortality is unknown.

### Methods

We use a national dataset from India to compare districts with high access to electricity (>90% of households) to districts with middle (50–90%) and low (<50%) access to electricity and estimate the effect of lack of electricity access on all-cause premature mortality.

### Results

In 2014, out of 597 districts in India, 174 districts had high access, 228 had middle access, and 195 had low access to electricity. When compared to districts with high access, districts with low access had higher rates of age-standardized premature mortality in both women (2.09, 95% CI: 1.43–2.74) and men (0.99, 0.10–1.87). Similarly, these districts had higher rates of conditional probability of premature death in both women (9.16, 6.19–12.13) and men (4.04, 0.77–7.30). Middle access districts had higher rates of age-standardized premature mortality and premature death in women, but not men. The total excess deaths attributable to reduced electricity access were 444,225 (45,195 in middle access districts and 399,030 in low access districts). In low access districts, the proportion of premature adult deaths attributable to low electricity access was 21.3% (14.4%– 28.1%) in women and 7.9% (1.5%– 14.3%) in men.

**Funding:** The author(s) received no specific funding for this work.

**Competing interests:** The authors have declared that no competing interests exist.

## Conclusion

Poor access to electricity is associated with nearly half a million premature adult deaths. One out of five premature deaths in adult women were linked to low electricity access making it a major social determinant of health.

## Introduction

Energy use is closely associated with social and economic development of a society [1]. Electricity is a central component of modern energy, steadily displacing non-electric forms of energy over time [2]. As societies aim to phase out fossil fuels in the face of climate change, access to clean electricity will be paramount to human development. Therefore, it is important to understand the relationship between electricity access and health outcomes.

India proposes an ambitious goal of becoming a global economic powerhouse by 2030, largely on the backbone of its relatively young population [3]. Efforts to preserve this young workforce involves combating the numerous individual socio-economic and environmental deprivations that impact premature mortality [4–7]. Access to reliable electricity has been linked to economic growth, leading successive Indian governments over the past two decades to take large strides in improving access to electricity from ~85% in 2014 to ~97% in 2020 [8,9]. However, there appears to be persistent regional variation in access to electricity, its reliability, and satisfaction among users [10]. Moreover, the health implications of expanded electricity access is unclear, particularly given that a lot of electricity is currently generated by fossil fuels. Additionally, there seems to be contradictory data regarding the impact of electrification on the health of populations living in low-and-middle income countries, largely driven by analyses using outcome measures that are surrogate to mortality. Hence, it is important to understand the impact of these governmental efforts to improve electrification on premature mortality of Indians.

Increased electricity access has been linked with improved respiratory health, reduced household medical expenditures, and improved nutrition in children [11]. Electrification has also been found to be associated with lower rates of maternal and child mortality through multiple potential mechanisms such as more hygienic methods of childbirth, food preparation, and water consumption [12]. On the contrary, household electrification has also been associated with an increased risk of malaria transmission, likely by attracting malaria vectors to light sources and allowing people to spend more time outdoors after sunset [13]. These contradictory reports of the impact of electrification on health warrants the need for defining the direct association between access to electricity and premature mortality in a nationwide sample of adults [11].

In 2014, Ram et al. reported that nearly two-fifths of India's men and one-third of women had >50% and >40% probability of dying before the age of 70 years, respectively. Importantly, they identified a large district-level survival gap with the highest premature mortality rates in east India and the lowest in west India [14]. Numerous infectious and non-communicable diseases accounted for only 60% of the gap with the remaining unexplained. With access to reliable electricity being one of the indicators of both socio-economic and environmental sustainability development, we sought to understand the effect of lack of electricity access on premature mortality and discuss potential pathways of a causal effect.

## Materials & methods

### Data sources

Premature mortality was defined as the death of an individual aged 15 years and above who died before reaching the age of 70 years. Based on the 2011 census, the country of India was divided into 640 districts for which district-level estimates were derived from various publicly available data sources. Census data was used to obtain information on the use of electricity for household lighting purposes, female literacy rate, female employment rate, district-level percentage of urban population, and other household amenities, including the use of polluting cooking fuels and access to a treated water source as the main source of potable water [15].

District-level data on annual fine particulate matter ($PM_{2.5}$) concentrations were obtained from a global database distributed by Dalhousie University, which estimated $PM_{2.5}$ concentrations by applying Geographically Weighted Regression (GWR) to data from satellites, models, and monitors [16].

District populations, age-standardized premature mortality rates, and conditional probabilities of premature death in 2014 between the ages of 15–69 years were obtained from the analysis performed by Ram et al. who used nationally representative data from various sources to derive internally consistent estimates [14]. Ram et al. classified the 12 smaller states and union territories (containing 39 districts) as single districts themselves. Hence, to align with the mortality data, the total district count included in our study was reduced from 640 to 597.

### Outcome measures

Two separate outcome measures were included in our analysis. District-level age-standardized premature mortality rate estimated in 2014 which is a point estimate. Conditional probability of premature death was defined as the probability that an individual aged 15 years would die before reaching the age of 70 years. Both outcome measures were derived through rigorous methods employed by Ram et al. and made publicly available [14].

### Statistical analysis

Districts were categorized into three groups based on the proportion of households with access to electricity for lighting purposes: >90% (highest access and reference category), 50–90% (middle access) and <50% (low access). We estimated the mortality effects of living in a district with reduced electricity access (middle access and low access districts) using multivariable linear regression models, with district level age-standardized premature mortality rates as the outcome measure in one set of models and conditional probability of premature death as the outcome measure in a separate set of models. Models were developed separately for men and women. To account for confounding of a variety of socio-ecological variables that could confound our exposure-outcome association, we adjusted for the following district-level variables: the total population of men and women in the district, percentage of urban population, female literacy, female employment, households using polluting cooking fuels, access to treated drinking water, and $PM_{2.5}$ concentrations in 2014. The actual number of excess premature deaths attributable to reduced electricity access was estimated using the coefficient of age-standardized premature mortality rates derived from the regression models and the total population living in districts with reduced electricity access. Within reduced electricity access districts, we also estimated the proportion of premature deaths attributable to reduced electricity access using the coefficient of conditional probability of death from the regression model and the mean conditional probability of death in those districts.

Scatterplots were used to visualize the relationship between age-standardized premature mortality rates and electricity access rates in districts. Pearson correlation coefficients were used to measure the strengths of correlation. Lastly, using similar linear regression models, we estimated the change in premature mortality rates and probability of premature death, per 1-standard deviation (SD) of increased access to electricity. We replicated this analysis by replacing electricity access with other more established social determinants of health (increased access to treated water, increased female literacy, and decreased use of polluting cooking fuel indicating household air pollution) as the independent variable. The change in overall premature mortality rates and probability of premature death was compared to show the impact of improving access to electricity. Estimates were made separately for men and women. All analyses were performed using R software v3.5.2 and STATA 16.1 with a two-way significance level of <0.05.

## Results

Baseline characteristics of districts, categorized by electricity access rates, are shown in Table 1. A total of 174 districts had >90% of its households with access to electricity, 228 districts had access rates between 50–90% and, 195 districts had <50% of its households with access to electricity (Fig 1). Nearly one-third of the population lived in low electricity access districts. Compared to districts with high electricity access, districts with low access were more rural, had lower rates of female literacy and employment, lower usage of treated drinking water, a greater proportion of households using polluting cooking fuels, and higher annual PM$_{2.5}$ levels.

### Premature mortality estimates for women and men living in districts with reduced (middle and low) electricity access

Table 2 shows that when compared to districts with high electricity access, middle access districts demonstrated a higher rate of age-standardized premature mortality among women

**Table 1. Baseline characteristics of districts, categorized by electricity coverage rates.**

| | Highest Access Districts (>90% of households) (n = 174) | Middle Access Districts (50–90% of households) (n = 228) | Lowest Access Districts (<50% of households) (n = 195) |
|---|---|---|---|
| Total population | 278,062,092 (33.4) | 292,725,324 (35.2) | 261,667,575 (31.4) |
| Women | 136,598,004 (49.1) | 144,258,108 (49.2) | 127,252,515 (48.6) |
| Urban population (%) | 46.1 ± 26.3 | 26.6 ± 17.1 | 12.6 ± 7.0 |
| Women who are literate (%) | 64.9 ± 9.9 | 54.1 ± 10.9 | 46.1 ± 9.4 |
| Women who work (%) | 18.1 ± 8.8 | 19.4 ± 9.8 | 10.9 ± 5.8 |
| District-level households using treated drinking water (%) | 56.3 ± 68.7 | 27.2 ± 16.7 | 9.6 ± 8.5 |
| District-level households using polluting cooking fuels (%) | 52.3 ± 19.2 | 76.8 ± 12.4 | 89.3 ± 5.5 |
| Annual district PM$_{2.5}$ level | 44.2 ± 22.3 | 44.6 ± 18.3 | 62.2 ± 23.0 |
| Age-standardized Death Rate* (Women) | 5.04 ± 1.67 | 5.95 ± 2.12 | 7.32 ± 2.18 |
| Age-standardized Death Rate* (Men) | 8.27 ± 2.74 | 8.67 ± 2.61 | 9.50 ± 3.02 |
| Conditional Probability of Death (%) (Women) | 32.7 ± 8.6 | 36.8 ± 9.4 | 43.1 ± 9.2 |
| Conditional Probability of Death (%) (Men) | 45.8 ± 10.3 | 47.8 ± 10.1 | 51.0 ± 10.5 |

*Expressed as rate per 1000 persons.

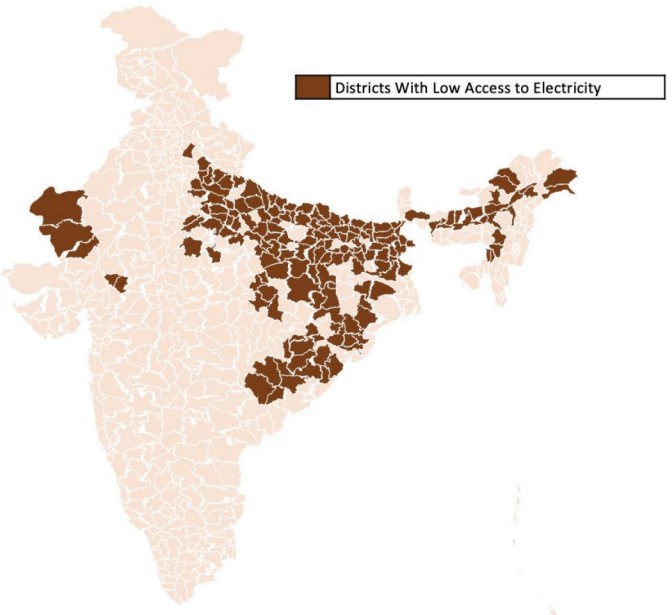

Reprinted from Gramener under a CC BY license, with permission from Gramener:
Insights as Data Stories, original copyright 2018.

**Fig 1. Map of India showing districts with <50% of its households having access to electricity.**

(0.55, 95% CI 0.06 to 1.03, p 0.03), but not among men (-0.23, -0.88 to -0.43, p 0.5). Similarly, there was an increased rate of conditional probability of premature death in women (2.41, 95% CI 0.22, 4.60, p 0.03) but not in men (-0.41, 95% CI -2.82 to 2.01, p 0.74). On average, 45,195 (79,342 in women and -34,147 in men) excess premature deaths were attributable to middle range access to electricity. Compared to high access districts, those with low access had higher rates of age-standardized premature mortality in both women (2.09, 95% CI 1.43 to 2.74, p <0.001) and men (0.99, 95% CI 0.10 to 1.87, p 0.03), and a higher conditional probability of

**Table 2. Effect of low electricity access on mortality rates and probability of premature death.**

| Variable | Middle Access District (95% CI, p-value) | Excess Deaths (N) | Low Access Districts (95% CI, p-value) | Excess Deaths (N) |
|---|---|---|---|---|
| Age-standardized premature mortality rate* (women) | 0.55 (0.06–1.03, 0.03) | 79,342 (8,656–148,586) | 2.09 (1.43–2.74, <0.001) | 265,959 (181,972–348,673) |
| Age-standardized premature mortality rate* (men) | -0.23 (-0.88–0.43, 0.50) | -34,147 (-130,651–63,841) | 0.99 (0.10–1.87, 0.03) | 133,071 (13,442–251,356) |
| Conditional probability of premature death (women) | 2.41 (0.22–4.60, 0.03) | NA | 9.16 (6.19–12.13, <0.001) | NA |
| Conditional probability of premature death (men) | -0.41 (-2.82–2.01, 0.74) | NA | 4.04 (0.77–7.30, 0.02) | NA |

*Expressed as per 1000 persons.

All models adjusted for percentage of district population that is urban, percentage of women in the district who are literate, total population of men and women in the district, percentage of females in the district who work, proportion of households in the district using polluting cooking fuels, proportion having access to a treated water, and district $PM_{2.5}$ levels.

premature death in both women (9.16, 95% CI 6.19 to 12.13, p <0.001) and men (4.04, 95% CI 0.77 to 7.30, p 0.02). On average, 399,030 excess deaths (265,959 women and 133,071 men) were attributable to the low access of electricity. The proportion of premature adult deaths attributable to reduced electricity access was 21.3% (14.4%– 28.1%) among women and 7.9% (1.5% to 14.3%) among men.

### Relationship between access to electricity and premature mortality

Fig 2 shows a linear fitted plot of age-standardized premature mortality rates (women and men) and electricity coverage rates, by district. There is an inverse relationship between premature mortality rates and district electricity access rates, slightly stronger for women (p = -0.41, p <0.0001) than men (p = -0.16, p = 0.0001). Fig 3 shows a linear fitted plot of conditional probability of premature death (women and men) and electricity access rates, by district. There is again an inverse relationship between the two variables, slightly stronger for women (p = -0.42, p <0.0001) than men (p = -0.17, p <0.0001).

### Comparison of electricity access with other social determinants of health

Table 3 shows the change in premature mortality rates and probability of premature death, per 1-SD increase in electricity coverage, in comparison with other established social determinants

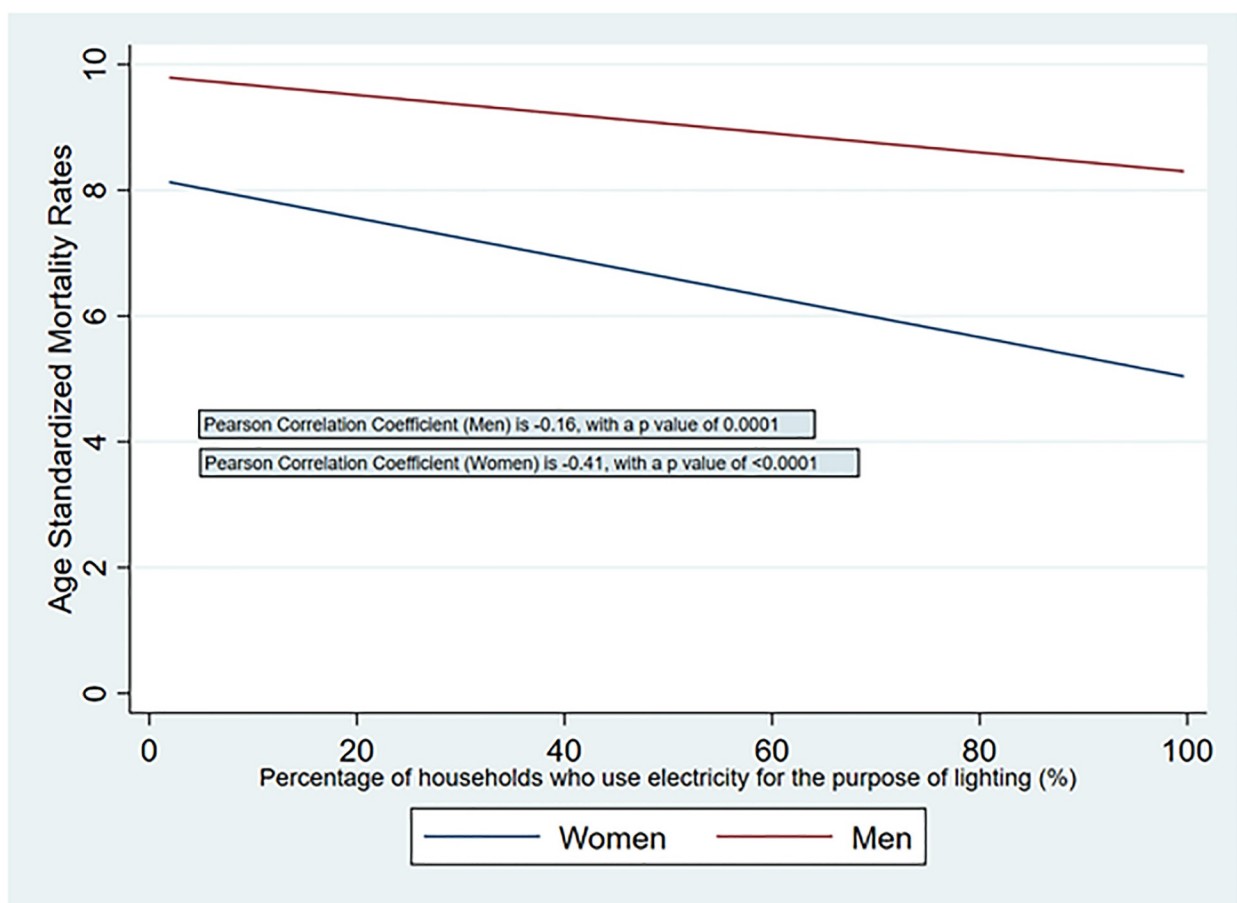

**Fig 2. Relationship between district-level age-standardized premature mortality rates (women and men) with access to electricity.**

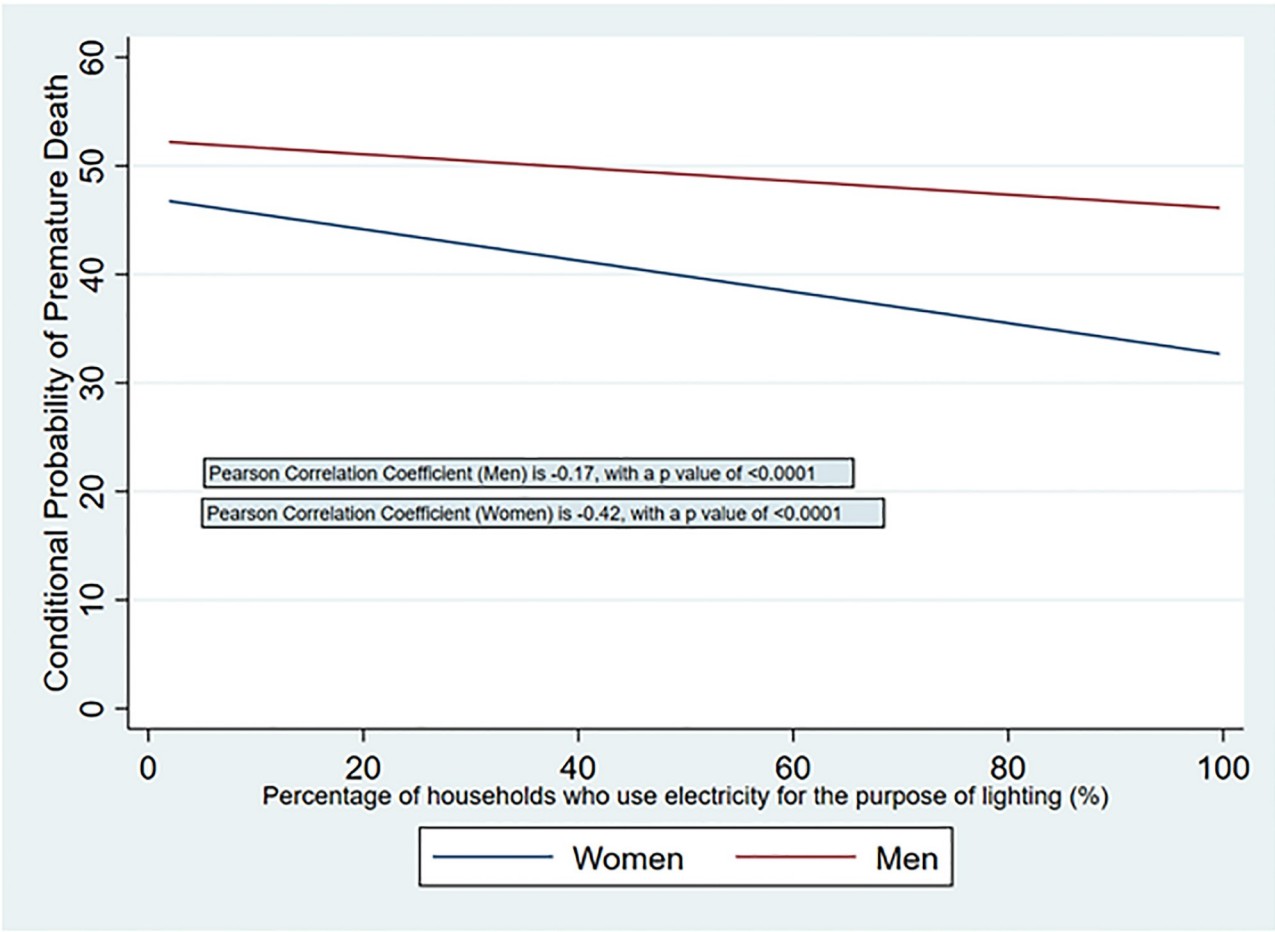

**Fig 3. Relationship between district-level conditional probability of death (women and men) with access to electricity.**

of health (increased literacy rates, reduced use of polluting cooking fuels, and increased access to treated water source). Access to electricity has a similar or stronger association with premature mortality and probability of premature death, as compared with other key social determinants of health.

**Table 3. Change in mortality rates and probability of premature death in comparison with established social determinants of health.**

|  | ASMR- Women | ASMR- Men | CPOD- Women | CPOD- Men |
|---|---|---|---|---|
| Increased access to electricity | -0.88 (-1.11, -0.64) | -0.34 (-0.66, -0.01) | -4.10 (-5.15, -3.05) | -1.30 (-2.50, -0.10) |
| Increased access to treated water | -0.07 (-0.25, 0.12) | -0.04 (-0.29, 0.30) | -0.30 (-1.13, 0.53) | -0.13 (-1.08, 0.81) |
| Increase in female literacy | -0.03 (-0.24, 0.19) | -0.01 (-0.29, 0.30) | 0.07 (-0.90, 1.04) | 0.03 (-1.07, 1.13) |
| Decrease in use of polluting fuel | -0.33 (-0.64, -0.02) | -0.51 (-0.94, -0.09) | -1.79 (-3.17, -0.40) | -2.20 (-3.78, -0.62) |

ASMR–Age-standardized mortality rate; CPOD–Conditional probability of premature death.

Numbers are absolute change (95% CI). Absolute change is per 1-SD increase in electricity access (28.8%), per 1-SD increase in access to treated water (23.2%), per 1-SD increase in literate women (12.6%) and per 1-SD decrease in polluting fuel use for cooking purposes (19.8%). Models adjusted mutually and for number of men and women in district and proportion of urban population in district.

## Discussion

Using a publicly available national database from 2014, we assessed the association between access to electricity and all-cause premature adult mortality in India. We found that there is large regional variation in the access to electricity across the country with an inverse relationship to premature adult mortality among both Indian women and men. Compared to districts with high electricity access, those with reduced access demonstrated higher rates of premature mortality in 2014 with people living in these deprived districts facing a higher probability of premature death. Importantly, we found that electricity access is a major social determinant of health, with ~400,000 premature adult deaths in India being associated with low electricity access in 2014. Additionally, the distribution of these deaths is highly inequitable and seems to be concentrated within one third of districts in the country. Lastly, increasing access to electricity seems to have a larger impact in reducing premature adult mortality in India compared to other established social determinants of health.

The fragmented distribution of electricity access in India demonstrates that most low access districts are clustered in the eastern part of the country (Fig 1) which interestingly coincides with the regional variation in premature mortality described by Ram et al. highlighting access to electricity to be an important social determinant of health [14]. Although prior reports have consistently shown lower rates of premature mortality among women compared to men, our analysis suggests that increasing access to electricity may disproportionately benefit women. In India, there is a strong inverse correlation between the level of electrification and maternal mortality rate [17]. Women are often responsible for kerosene related tasks in the household and may suffer disproportionately from its impact. An observation in Chhattisgarh, India demonstrated that village electrification resulted in improving flexibility in time use by postponing dinner cooking by ~40 minutes. This was further accompanied by a 67% decrease in the use of kerosene. Additionally, several broader effects of electrification on gender norms have been described such as enhanced women's employment, reduced time spent on drudgery, and improved women's literacy rates [18]. Furthermore, access to communication systems, a key component of women's empowerment is critically dependent on access to reliable electricity [19].

Despite its multiple applications, electricity remains to be primarily used for lighting purposes. In the absence of electricity, kerosene is a commonly used source of lighting. Kerosene is a fossil fuel that has a wide range of deleterious impacts on health, including indoor air pollution, impaired lung function, increased risk of infectious diseases and an increased risk of poisonings and fires [20]. In addition to decreasing kerosene use, electricity access likely affects health through a wide range of mechanisms, as electricity is an enabling mechanism for numerous human systems that have an impact on health. Positive effects on food systems, water systems, lighting and cooling, communication, and healthcare systems likely account for a large share of electricity's role in enhancing health [21]. Additionally, through increased business activities in the community, electricity access enhances both agriculture and non-agriculture income, yet another important determinant of premature mortality [21,22]. Fig 4 shows the potential pathways by which reliable electricity coverage can improve adult premature mortality rates [11].

Similar to female literacy, access to treated drinking water, and clean cooking fuels, access to clean and reliable electricity needs to be recognized as a major social determinant of health. In fact, improving these related social determinants of health is partly contingent on reliable electricity access. Electricity is required for the operation of large-scale water treatment facilities, electric cooking, use of clean cookstoves (forced draft), and has been consistently associated with higher literacy rates [23,24].

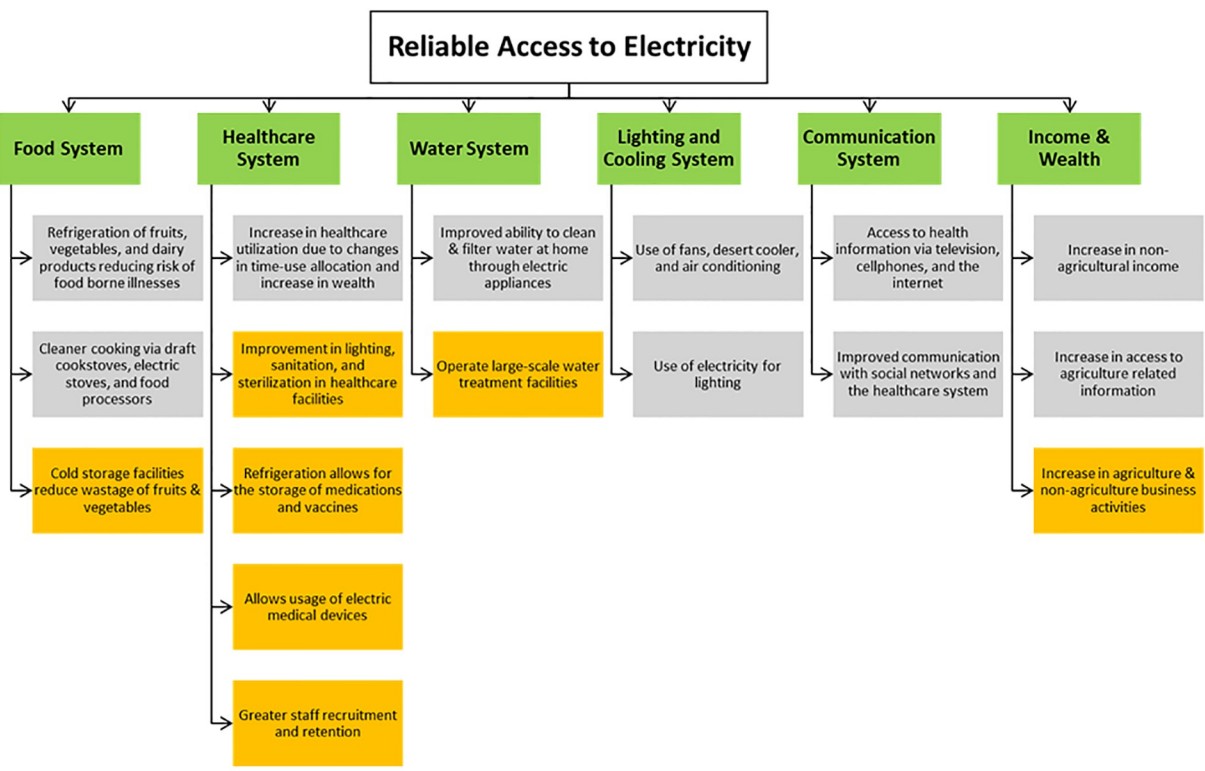

**Fig 4. Conceptual model showing potential pathways through which reliable electricity coverage can improve adult premature mortality rates.**

Realizing the full benefits of electricity will require, along with 100% access, a zero carbon, reliable supply. Coal-fired power plants account for 51% of India's installed power capacity in 2022 [25]. The use of coal for electricity generation has been associated with a wide range of detrimental health and environmental impacts, including ~50,000 premature adult deaths annually [26]. In addition, coal is a key driver of climate change, which is already leading to substantial adverse health impacts in India. A clean electric grid, built on solar, wind and other renewables, along with hydro and nuclear for baseload capacity, will lead to a sustainable electric future that maximizes health gains.

The national proportion of households that used electricity for lighting purposes in the 2011 census was 67.3%. The 2021 decadal census, which would allow this metric to be updated, has still not been completed [27]. In its absence, sample-based surveys such as the National Family Health Survey (NFHS) have shown that electricity coverage rates have improved significantly since the 2011 census. While NFHS-4 (2015–16) showed that the population living in households with electricity was 88%, this proportion further increased to 96.8% in NFHS-5 (2019–21) [28]. However, electricity reliability (hours of electricity supply in a day and voltage stability) continues to be variable across India [10]. Moreover, COVID-19 has reversed progress in electricity access, as the number of people with a low socioeconomic status in India is

estimated to have more than doubled in just a year (from 60 million to 134 million) [29]. The impact of this on access to reliable electricity remains unclear.

Our findings must be interpreted in the context of several limitations. First, given the observational nature of the study, residual confounding cannot be excluded. Second, we performed a cross-sectional analysis to assess the association between electricity access and premature death and cannot assume causality. Third, we do not measure the effect of electricity reliability (duration of supply in a day, power outages, voltage fluctuations), which will likely be important to fully realize the beneficial health effects of electricity access. Fourth, given that India's electricity coverage has increased since the 2011 census, the estimates do not reflect the current burden of mortality. However, given the postponement of the 2021 census, more recent data is not available.

In conclusion, we show that ~400,000 of India's premature adult deaths were attributable to low electricity coverage in 2014, which were concentrated in one third of Indian districts. Along with increasing electricity access, ensuring a reliable, zero-carbon supply will be crucial to realizing the full potential of electrification to enhance human health.

## Supporting information

**S1 Dataset.**
(XLS)

## Acknowledgments

We appreciate the assistance provided by Aviraag Vijayaprakash, MBBS and Abilash Muralidharan, MBBS in the completion of this data analysis.

## Author Contributions

**Conceptualization:** Vittal Hejjaji, Dweep Barbhaya, Aditya K. Khetan.

**Data curation:** Vittal Hejjaji, Dweep Barbhaya, Amirarsalan Rahimian, Aishwarya Yamparala, Shreyas Yakkali.

**Formal analysis:** Vittal Hejjaji, Amirarsalan Rahimian, Aditya K. Khetan.

**Methodology:** Vittal Hejjaji.

**Project administration:** Vittal Hejjaji, Dweep Barbhaya, Amirarsalan Rahimian, Shreyas Yakkali.

**Resources:** Amirarsalan Rahimian, Aishwarya Yamparala.

**Software:** Vittal Hejjaji, Dweep Barbhaya, Amirarsalan Rahimian, Aishwarya Yamparala, Shreyas Yakkali.

**Visualization:** Vittal Hejjaji, Dweep Barbhaya.

**Writing – original draft:** Vittal Hejjaji, Aditya K. Khetan.

**Writing – review & editing:** Vittal Hejjaji, Dweep Barbhaya, Amirarsalan Rahimian, Aishwarya Yamparala, Shreyas Yakkali, Aditya K. Khetan.

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
