## [Decision Letter · Decision Letter 0]

12 Sep 2023

PONE-D-23-25292The Burden of Premature Adult Mortality Associated with Lack of Access to Electricity in IndiaPLOS ONE

Dear Dr. Khetan,

Thank you for submitting your manuscript to PLOS ONE. After careful consideration, we feel that it has merit but does not fully meet PLOS ONE’s publication criteria as it currently stands. Therefore, we invite you to submit a revised version of the manuscript that addresses the points raised during the review process.

We look forward to receiving your revised manuscript.

Kind regards,

Ranjit Kumar Dehury

Academic Editor

PLOS ONE

Journal Requirements:

2. We note that Figure 1 in your submission contain map/satellite images which may be copyrighted. All PLOS content is published under the Creative Commons Attribution License (CC BY 4.0), which means that the manuscript, images, and Supporting Information files will be freely available online, and any third party is permitted to access, download, copy, distribute, and use these materials in any way, even commercially, with proper attribution. For these reasons, we cannot publish previously copyrighted maps or satellite images created using proprietary data, such as Google software (Google Maps, Street View, and Earth). For more information, see our copyright guidelines: http://journals.plos.org/plosone/s/licenses-and-copyright.

Additional Editor Comments:

Dear authors,

The paper needs major revision by incorporating the reviewers comment. Hence, I would recommend for major revision.

With regards,

Ranjit

Reviewers' comments:

Reviewer's Responses to Questions

**Comments to the Author**

1. Is the manuscript technically sound, and do the data support the conclusions?

Reviewer #1: Yes

Reviewer #2: No

Reviewer #3: Yes

2. Has the statistical analysis been performed appropriately and rigorously? 

Reviewer #1: Yes

Reviewer #2: No

Reviewer #3: Yes

3. Have the authors made all data underlying the findings in their manuscript fully available?

Reviewer #1: Yes

Reviewer #2: No

Reviewer #3: Yes

4. Is the manuscript presented in an intelligible fashion and written in standard English?

Reviewer #1: No

Reviewer #2: No

Reviewer #3: Yes

5. Review Comments to the Author

Reviewer #1: An interesting subject has been notified in this manuscript; however, data has been collected in 2014 (9 years ago).

It is recommended to use the verbs in past tense, not present, in "Methods" and also in the last paragraph of "Introduction" section (we performed).

"Premature Death" should be defined, before it has been reported in "Results".

Please check the "Keywords" according to main heading terms of the MeSH.

Fig 4 has been inserted in text with citation to this reference: (Irwin BR, Hoxha K, Grépin KA. Conceptualising the effect of access to electricity on health in low259 and middle-income countries: A systematic review). Please get a written permission from the corresponding author for using this figure.

Reviewer #2: The author did not provide sufficient literature to highlight the issues. More literature is required to find the research gaps.

What are the other factors associated with Pre-mature adult death in India? What are the morbid conditions if lack of electricity access is the only factor? More evidence is required to establish your arguments.

What are the variables abstract from DLHS or Census, 2011-India? The author should give the details about the process of data accessibility. What is the procedure followed to analyse the data?

The morbid condition associated with low electricity access is missing from the result section. The author is required to follow the methodology in a rigorous way.

The author should highlight the key inference of the present study, how this study proved the lack of electric access leads to adult mortality.

Reviewer #3: The paper is well-written and succinct, but it would benefit from a mention of the methodology used and the data sources employed in the study. Additionally, specifying the geographic representation of the chosen districts and discussing policy implications could enhance the paper's completeness. In the paper, it's essential to provide more details on the methodology, acknowledge data limitations and potential biases, and address the issue of causality. Exploring variation within low-access districts and offering insights into the gender disparities observed would strengthen the paper's analysis.

6. PLOS authors have the option to publish the peer review history of their article (what does this mean?). If published, this will include your full peer review and any attached files.

Reviewer #1: No

Reviewer #2: No

Reviewer #3: **Yes: **Abhishek Dondapati

---

## [Author Response · Author response to Decision Letter 0]

25 Oct 2023

The response to reviewers has been attached in a letter. For editor comments, 

1) We have formatted the manuscript to meet PLOS ONE's style requirements. 

2) We have obtained permission for Figure 1, which has been attached as an 'other' file. This permission is also reflected in the figure itself. 

3) We have added a caption for supporting information files in the manuscript.

---

## [Decision Letter · Decision Letter 1]

21 Nov 2023

PONE-D-23-25292R1The Burden of Premature Adult Mortality Associated with Lack of Access to Electricity in IndiaPLOS ONE

Dear Dr. Khetan,

Thank you for submitting your manuscript to PLOS ONE. After careful consideration, we feel that it has merit but does not fully meet PLOS ONE’s publication criteria as it currently stands. Therefore, we invite you to submit a revised version of the manuscript that addresses the points raised during the review process.

**ACADEMIC EDITOR: **

Dear authors

Please make a through revision of the article by incorporating the comments of the reviewers, including the reviewer rejected the article. 

With regards,

Ranjit

We look forward to receiving your revised manuscript.

Kind regards,

Ranjit Kumar Dehury

Academic Editor

PLOS ONE

Additional Editor Comments:

Dear authors

Please make a through revision of the article by incorporating the comments of the reviewers, including the reviewer rejected the article.

With regards,

Ranjit

Reviewers' comments:

Reviewer's Responses to Questions

**Comments to the Author**

1. If the authors have adequately addressed your comments raised in a previous round of review and you feel that this manuscript is now acceptable for publication, you may indicate that here to bypass the “Comments to the Author” section, enter your conflict of interest statement in the “Confidential to Editor” section, and submit your "Accept" recommendation.

Reviewer #1: All comments have been addressed

2. Is the manuscript technically sound, and do the data support the conclusions?

Reviewer #1: Yes

3. Has the statistical analysis been performed appropriately and rigorously? 

Reviewer #1: Yes

4. Have the authors made all data underlying the findings in their manuscript fully available?

Reviewer #1: Yes

5. Is the manuscript presented in an intelligible fashion and written in standard English?

Reviewer #1: Yes

6. Review Comments to the Author

Reviewer #1: Thanks for consideration of the recommended points.

My comments have been notified in the revised version of the article.

7. PLOS authors have the option to publish the peer review history of their article (what does this mean?). If published, this will include your full peer review and any attached files.

Reviewer #1: No

---

## [Author Response · Author response to Decision Letter 1]

3 Jan 2024

1. The authors have addressed most of the points raised by the reviewers in the earlier version. The article is substantially improved.

2. The introduction section should have a paragraph which needs to highlight the electricity situation in India and the concerned study areas.

RESPONSE: We thank the reviewers for considering our previous modifications and providing us with further feedback in improving our manuscript. As suggested by the reviewer, we have now restructured our ‘Introduction’ section and included a new paragraph that clearly describes the rationale behind our aim to understand the association between access to electricity and premature death among Indians. Changes have been made to page 2 as described below: 

‘India proposes an ambitious goal of becoming a global economic powerhouse by 2030, largely on the backbone of its relatively young population.(3) Efforts to preserve this young workforce involves combating the numerous individual socio-economic and environmental deprivations that impact premature mortality.(4-7) Access to reliable electricity has been linked to economic growth, leading successive Indian governments over the past two decades to take large strides in improving access to electricity from ~85% in 2014 to ~97% in 2020.(8, 9) However, there appears to be persistent regional variation in access to electricity, its reliability, and satisfaction among users.(10) Moreover, the health implications of expanded electricity access is unclear, particularly given that a lot of electricity is currently generated by fossil fuels. Additionally, there seems to be contradictory data regarding the impact of electrification on the health of populations living in low-and-middle income countries, largely driven by analyses using outcome measures that are surrogate to mortality. Hence, it is important to understand the impact of these governmental efforts to improve electrification on premature mortality of Indians.’’

3. Why the issue is so important to be enquired need to be justifies in the introduction section.

RESPONSE: The restructuring of our Introduction section and the inclusion of additional information as mentioned in Comment 1 enhances the justification for studying the issue of electricity. 

4. Further, justification of national database from 2014 is needed as it is one decade old and a new government is in place In India.

5. Why not recent data needs to be explained.

RESPONSE: A key part of our analysis was the requirement of data from the Indian Census. The most recent census data available is from 2011 with the 2021 census being postponed. This is the primary reason for using a database from 2014, which is based on the 2011 census. However, given the goal of our analysis is to merely understand the link between electricity access and premature death, we find that a cross sectional analysis at any time point should provide the necessary information. To ensure completeness, we have mentioned this to be a limitation in the Discussion section on page 11. 

‘…Fourth, given that India’s electricity coverage has increased since the 2011 census, the estimates do not reflect the current burden of mortality. However, given the postponement of the 2021 census, more recent data is not available.’

6. Discussion can compare some of the studies of global south if data is available.

7. Overall the study provide an important message which needs to be presented in a story telling manner.

RESPONSE: Data from countries in the global south with regards to this topic is unfortunately limited. References 11-13, which are mentioned and explained in the introduction, summarize the prior literature on this issue. The paucity of data globally on the relationship between premature mortality and access to electricity was the primary motive behind preparing this manuscript.

---

## [Editor Report · Decision Letter 2]

8 Jan 2024

The Burden of Premature Adult Mortality Associated with Lack of Access to Electricity in India

PONE-D-23-25292R2

Dear Dr. Khetan,

We’re pleased to inform you that your manuscript has been judged scientifically suitable for publication and will be formally accepted for publication once it meets all outstanding technical requirements.

Kind regards,

Ranjit Kumar Dehury

Academic Editor

PLOS ONE

Additional Editor Comments (optional):

Dear author,

The article has been improved a lot by incorporating the reviewers comments. Hence, the manuscript is accepted for publication. However the authors are advised to comply with technicalities of the journal.

With regards,

Ranjit
---

## [Editor Report · Acceptance letter]

2 Mar 2024

PONE-D-23-25292R2 

PLOS ONE

Dear Dr. Khetan, 

I'm pleased to inform you that your manuscript has been deemed suitable for publication in PLOS ONE. Congratulations! Your manuscript is now being handed over to our production team.

Kind regards, 

on behalf of

Dr. Ranjit Kumar Dehury 

Academic Editor

PLOS ONE